



# Ch3MS-RF: A Random Forest Model for Chemical Characterization and Improved Quantification of Unidentified Atmospheric Organics
Detected by Chromatography-Mass Spectrometry Techniques

Emily B. Franklin[1], Lindsay D. Yee[2], Bernard Aumont[3], Robert J. Weber[2], Paul Grigas[4], Allen Goldstein[1,2]

[1]Department of Civil and Environmental Engineering, University of California Berkeley, Berkeley, 94720, USA
[2]Department of Environmental Science, Policy and Management, University of California Berkeley, Berkeley, 94720, USA
[3] Univ Paris Est Creteil and Université de Paris, CNRS, LISA, F-94010 Créteil, France
[4]Department of Industrial Engineering and Operations Research, University of California Berkeley, Berkeley, 94720, USA

*Correspondence to*: Emily B. Franklin (barnes_emily@berkeley.edu), Allen  H. Goldstein (ahg@berkeley.edu)



**Abstract.** The chemical composition of ambient organic aerosols plays a critical role in driving their climate and health relevant properties and holds important clues to the sources and formation mechanisms of secondary aerosol material. In most ambient atmospheric environments, this composition remains incompletely characterized, with the number of identifiable

species consistently outnumbered by those that have no mass spectral matches in the literature or NIST/NIH/EPA mass spectral databases, making them nearly impossible to definitively identify. This creates significant challenges in utilizing the full analytical capabilities of techniques which separate and generate spectra for complex environmental samples. In this work, we develop the use of machine learning techniques to quantify and characterize novel, or unidentifiable, organic material. This work introduces Ch3MS-RF (Chemical Characterization by Chromatography-Mass Spec Random Forest Modelling), an open-

source R-based software tool for efficient machine-learning enabled characterization of compounds separated in chromatography-mass spec applications but not identifiable by comparison to mass spectral databases. A random forest model is trained and tested on a known 130 component representative external standard to predict the response factors of novel environmental organics based on position in volatility-polarity space and mass spectrum, enabling reproducible, efficient, and optimized quantification of novel environmental species. Quantification accuracy on a reserved 20% test set randomly split

from the external standard compound list indicate that random forest modelling significantly outperforms the commonly used methods in both precision and accuracy, with a median response factor % error of -2% for modelled response factors compared to >15% for typically used proxy assignment-based methods. Chemical properties modelling, evaluated on the same reserved 20% test set as well as an extrapolation set of species identified in ambient organic aerosol samples collected in the amazon rainforest, also demonstrates robust performance. Extrapolation set property prediction mean average errors for carbon

number, oxygen to carbon ratio (O:C), average carbon oxidation state ($\overline{OS_c}$), and vapor pressure are 1.8, 0.15, 0.25, and 1.0 (log(atm)), respectively. Extrapolation set Out-of-Sample $R^2$ for all properties modelled are above 0.75, with the exception of vapor pressure. While predictive performance for vapor pressure is less robust compared to the other chemical properties modelled, random forest-based modelling was significantly more accurate than other commonly used methods of vapor pressure prediction, decreasing mean average vapor pressure prediction error to 0.24 (log(atm)) from 0.55 (log(atm))

(chromatography-based vapor pressure prediction) and 1.2 (log(atm)) (chemical formula-based vapor pressure prediction). The random forest model significantly advances untargeted analysis of the full scope of chemical speciation yielded by GCxGC-MS techniques and can be applied to GC-MS as well. It enables accurate estimation of key chemical properties commonly utilized in the atmospheric chemistry community, which may be used to more efficiently identify important tracers for further individual analysis and to characterize compound populations uniquely formed under specific ambient conditions.

**1   Introduction**

Organic aerosols play a critical role in global radiative forcing and regional aerosol-attributable public health concerns, making up a significant (20-90%) fraction of fine particulate matter around the globe (Jimenez et al., 2009). This organic material is highly complex in terms of chemical composition and constantly changing; Goldstein and





Galbally, 2007 estimates the number of gas and aerosol-phase atmospheric organic constituents to lie in the millions, while Ditto et al., 2018 reports molecular level variability of 60-80% between consecutive samples collected at fixed sites for samples comprised of high thousands of resolvable species. While there has been significant progress towards achieving mass closure of atmospheric reactive carbon using an ensemble of both bulk and speciated measurement techniques over the past two decades, speciated and isomer identified mass closure remains challenging (Heald et al., 2010; Hunter et al., 2017; Isaacman-Vanwertz et al., 2018). A comprehensive review of the challenges and utility of different levels of molecular identification, Nozière et al., 2015, compares the utility of many types of incomplete identification of atmospheric organic compounds, but defines that "An organic compound is fully identified only if its molecular structure is entirely known, including its isomeric and spatial (stereo) configuration." Important chemical information can be gleaned from formula-based identifications and bulk characterization, but isomer-specific identifications provide critical atmospheric chemistry-relevant insights. As described in Isaacman-Vanwertz and Aumont, 2021, different isomers of the same chemical formula vary over orders of magnitude in volatility and Henry's constant, and by a factor of 2 in reactivity with the hydroxyl radical, all critical properties for characterization of aerosol formation and properties. Isomer specific identifications also play a crucial role in elucidation of important chemical reaction mechanisms.

Gas chromatography coupled with electron ionization mass spectrometry (GC-MS) is a commonly utilized technique for isomer specific speciation of atmospheric constituents. Observed ambient species may be matched to authentic standards or mass spectral database entries by both retention index (chromatographic elution time relative to that of a series of alkanes) and mass spectrum. Two dimensional gas chromatography (GCxGC-MS), a methodologically similar technique which achieves advanced separation by passing compounds through multiple GC columns configured for different chemical properties, increases the scope of isomer-specific identification by separating species that would coelute in single dimension GC-MS applications (Goldstein et al., 2008; Worton et al., 2011, 2017). However, a significant challenge of fully utilizing the data from these techniques is the novelty and diversity of the atmospheric constituents; most observed organic species have never been synthesized and are not in any mass spectral library and are therefore not directly identifiable from GC-MS or GCxGC-MS techniques. Although the size of mass spectral libraries are rapidly increasing, with ~30,000 new compounds added to the NIST/EPA/NIH mass spectral database between the 2011 and 2014 versions (bringing the number of compounds catalogued in NIST14 EI library to ~250,000), the numbers of identifiable constituents in typical atmospheric samples remain low (Vinaixa et al., 2016). As described in Hamilton et al., 2004, in an urban aerosol sample analyzed by GCxGC-MS, of >10,000 unique observed species, fewer than 2% were identifiable from authentic standard or mass spectral matching. Low numbers of matched relative to novel ambient species persist; Worton et al., 2017 finds that fewer than 35% of ~500 compounds isolated from aerosol samples collected at a forested site match mass spectral database entries, while this work (as later described) finds that fewer than 10% of ~1500 aerosol phase organic species can be matched to published spectra. As described in Worton et al., 2017, species that cannot be identified are often not included in GC-MS and GCxGC-MS-based analyses, meaning that the majority of



acquired data is not fully utilized. Note that in accordance with the definition of complete molecular identification previously quoted from Nozière et al., 2015, "unidentified" compounds are from here on defined as any species that is

not identifiable by comparison (on the basis of retention index and mass spectrum) to either authentic standards or mass spectral database entries of positively identified species. Pairing GC-EI-MS systems with complementary measurements such as chemical ionization (described in Bi et al., 2021) or switching to softer election ionization techniques (specifically through employing 14 ev vacuum ultraviolet rather than traditional 70 ev EI, intended to preserve sufficient precursor ion mass for formula identification, as described in Worton et al., 2017) can enable more separated but unidentified

compounds to be characterized by formula identification, even where isomer-specific identification remains elusive. That said, these instrumental configurations are rare, and fragmentation under 14 ev is still sufficiently significant to leave many species' formulae not identifiable and therefore still uncharacterized (Worton et al., 2017). Recent efforts to embrace a larger fraction of the full complexity of chemical information yielded by highly speciated organic aerosol measurements (on the scale of low to mid 100's of compounds) have categorized unidentified  species by likely source

groups and chemical families through time series correlations with known tracer species (Zhang et al., 2018) or by manual group assignments by individual researcher judgements based on mass spectral features (Liang et al., 2021). These methods are difficult to standardize and reproduce and become prohibitively inefficient when pushing towards the full chemical complexity of speciated observations produced from typical atmospheric samples, which extend into the low to mid thousands of species.

Quantification of unidentified compounds faces similar challenges.  Where possible, compounds in GC and GCxGC-MS are directly quantified by calibration curves of authentic standards, but direct quantifications are limited by standard expense and availability, even for species that can be positively identified. Compounds that cannot be directly quantified, both in GCxGC-MS and in GC-MS, are most commonly quantified by assigning quantification factors from compounds resolved closely in chromatographic space, compounds that are identified as sharing chemical structures, or some

interpolation of multiple nearby proxies (Hatch et al., 2015; Jen et al., 2019; Liang et al., 2021; Zhang et al., 2018). The errors associated with these assignments/choices are usually estimated from the range of quantification factors of close or chemically similar species and are assumed to be high (up to a factor of 2 depending on degree of certainty in assigning chemical class as described in Jen et al., 2019 and Liang et al., 2021). To our knowledge, this work presents the first quantitative error analysis of these techniques based on applying proxy quantification techniques to compounds with

known quantification factors.

Current manual characterization and quantification proxy-assignments are essentially an exercise of pattern recognition, as researchers use experience in analysing spectra and position in chromatographic space to categorize or otherwise characterize unidentifiable species. Given the scale of the novel compound characterization challenge (on the order of hundreds to thousands of species for a given sampling location using current methods), transitioning to automated

characterization methods will be necessary to keep up with data acquisition, and will offer co-benefits in increased reproducibility and reduced susceptibility to researcher biases. Decision tree-based machine learning methods including


gradient boosting and random forests have demonstrated robust performance in pattern recognition-based regression applications including nonlinear features across a wide range of fields (Bentéjac et al., 2021; Rokach, 2016). Random forests, a decision tree-based method which generates predictions based on a combination of diverse trees generated by

randomized feature selection and resampling on a training data set (Breiman, 2001), are particularly suited to this application and intended audience. They have demonstrated robust performance across a range of applications, including predictions of chemical properties (Whitmore et al., 2016) and do not require extensive hyperparameter tuning to achieve high performance (Bentéjac et al., 2021). In this work, we develop machine learning models, specifically based on the random forests methodology, that use  chromatographic and mass spectral feature inputs to predict a diverse suite of

chemical properties, including quantification factor in a TD-GCxGC-MS system, oxygen to carbon ratio (O:C), carbon number, average carbon oxidation state ($\overline{OS_c}$), and vapor pressure.  Coinciding with this manuscript, we have released a repository template including an Rmarkdown notebook (https://github.com/ebarnesey/Ch3MS-RF) that enables users with general atmospheric chemistry background, who do not necessarily have special expertise in machine learning data science applications, to tailor our analysis for their specific use cases.  As such, robust performance evaluation and ease

of applicability to a range of potential use cases are emphasized over extensive application-specific hyperparameter tuning.

In summary, this work aims to provide the GC-MS and GCxGC-MS atmospheric chemistry community with tools to achieve the following objectives:

1) Enable accurate chemical characterization of organic constituents separated in gas chromatographic space but not
145        necessarily published (in mass spectral databases)

2) Improve the quantification accuracy for species that cannot be directly calibrated using authentic standards

## 2    Instrumentation and Data

### 2.1    Calibration Curves Using an External Standard Mixture of Authentic Standards

A custom calibration standard mixture (referred to hereafter as "external standard") was created containing ~130 unique
authentic standards selected for maximal coverage of the compounds and compound classes typically observed in atmospheric regions with significant biogenic emissions, as well as influences from anthropogenic activities and biomass burning. The selection of these standard species was informed by previous work targeting similar sample types using the same instrumentation (Worton et al., 2011; Yee et al., 2018; Zhang et al., 2018), and covers species including sugars, PAH's, and both monoterpene and isoprene oxidation products. In addition to commercially available external standards, 6 sesquiterpene
oxidation products were custom synthesized by collaborators (as described in Bé et al., 2019) for expanded coverage of potentially important chemical tracers. The full list of standard components can be found in Table A1, and the standard property distribution in volatility-polarity space is illustrated in Figure 2. The standard was prepared from pure components immediately prior to sample analysis in 1:1 methanol:chloroform solution, replicating the methodology utilized in Zhang et al., 2018.



Standards were introduced to the instrument by injecting onto tissuquartz filter material to maximize consistency between filter

samples (organic aerosol also collected on tissuquartz filters) and calibration runs. At 5 points throughout sample analysis, 6-point calibration curves (5 loaded points and a zero point) were performed to determine the "quantification factors" (internal standard normalized signal/ng compound) of each external standard species. The internal standard, described in detail in section 2.3.1, is a solution of ~30 deuterated organics applied identically to all sample and calibration analysis runs to enable correction for instrument condition and matrix effects. For efficiency, outlier calibration points (significantly deviating from

the slope of other points in the quantification factor, which are often caused by coelution with a contaminant) were removed. A minimum of 3 calibration points above the zero point was maintained to ensure robust quantification factors.

## 2.2    GoAmazon Field Data

The ambient extrapolation data utilized in this work originates from the Green Ocean Amazon (GoAmazon) field campaign which was conducted in central Amazonia in 2014. This campaign and the collection of ambient filters for offline

analysis are described in detail in Martin et al., 2016, 2017 and Yee et al., 2018. Briefly, the campaign was conducted at a semi remote site occasionally downwind of the city of Manaus and periodically impacted by smoke from biomass burning. The campaign spanned two intensive operating periods, one during the Amazonian wet season (February through March) and one during the dry season (August through early October). Submicron aerosol samples were collected on tissuquartz filters (Pallflex), stored in pre-baked foil, double contained in sealed mylar bags, and frozen prior to analysis. The samples were

analysed by TD-GCxGC-EI-ToF-MS, as described below.

## 2.3    Instrumentation: TD-GCxGC-EI-ToF-MS

Both external standard species (during calibration runs) and GoAmazon filter samples were analysed by thermal desorption two-dimensional gas chromatography coupled with electron ionization time-of-flight mass spectrometry (TD-GCxGC-EI-ToF-MS, hereafter abbreviated GCxGC-MS). This instrumentation is described in detail in Goldstein et al., 2008

and Worton et al., 2011, and instrument specifics including sub-component models, column materials, and temperature settings are described in Franklin et al., 2021. For ambient filter samples, 0.4 cm$^2$ aliquots of filter material are directly introduced into the instrument. Standards are stored in solution and introduced by injection onto pre-baked quartz filter material. An internal standard (described in section 2.3.1) is applied on top of the sample or external standard filter aliquots immediately prior to analysis. Briefly, the instrument functions as follows: a thermal desorption oven heats filter material, causing analytes and

standards to evaporate into a flow of helium. The desorbed components are focused on a cooled inlet system (Gerstel CIS), which at the end of the thermal desorption cycle is rapidly heated to simultaneously release all organic species onto the head of the first column. Compounds are separated by both volatility and polarity by two gas chromatography columns in sequence, with the transition of compounds from the first to the second column modulated by a cryogenic focus and rapid thermal release system. Separated analytes are ionized by 70 eV electron ionization (EI) and detected by HR-ToF-MS (TOFWERK, EI-

HTOF), with a resolving power of 4000 acquired at 100 Hz. While the mass spectra produced by this technique are high



resolution, these high resolution mass spectra are converted to unit mass resolution spectra to increase the applicability of this technique to unit mass resolution techniques. The vertical (polar) axis of separation is extremely short relative to the horizontal (volatility) axis separation with a vertical stride length of 2.3 seconds compared to a retention time of ~ 1 hr for low volatility organics. As a result, GCxGC-MS deuterated alkane normalized retention indices are directly comparable to retention indices

(or, with a linear conversion to non-deuterated retention indices, kovats indices) in single dimension GC-MS applications. This instrument's volatility range spans approximately $C_{13}$-$C_{36}$ *n*-alkane volatility equivalents, covering the atmospherically important transition regime between IVOC (intermediate volatility organic carbon) and LVOC (low volatility organic carbon) species.

During the thermal desorption process, the carrier flow of helium is enriched with the derivatization agent MSTFA (n-

methyl-n-trimethylsilyl-trifluoro-acetamide). This silylating reagent replaces the active hydrogen of polar OH groups with a trimethylsilyl group, -Si(CH$_3$)$_3$, a process which significantly enhances the recovery of polar organics. This approach is critical to increase the scope and degree of oxygenation of species recovered by thermal desorption-gas chromatography techniques (Isaacman et al., 2014). However, it poses some challenges for data interpretation for diverse, complex, and novel chemical mixtures, because in the case of many polar species, the compound that is separated and detected by the GCxGC-MS

instrumentation has been chemically altered from the species that was collected. This can create challenges in compound identification, as not all species have published derivatized spectra, as well as challenges for mapping chemical properties onto the GCxGC-MS space, as the volatility-polarity distributions of derivatized compounds do not directly reflect their underivatized properties.

### 2.3.1   Internal Standard Normalization

Both filter samples and external standard impregnated filters (for calibration curves) were doped with a custom 23 component deuterated internal standard covering the full range of volatility sensitivity and a broad variety of functional group types immediately prior to analysis. The internal standard enables normalization for matrix effects, configuration of retention indices relative to the elution times of a deuterated alkane series, and normalization for instrument condition drift for improved consistency and quantification accuracy throughout intensive instrument use. In prior methods, the selection of internal

standard involved either1) assigning each analyte an internal standard nearest in chromatographic space (by retention times) or 2) manual assignment of analytes to their most chemically similar internal standards regardless of proximity in GCxGC space. Analyte signal would then be normalized (divided) by the signal of the selected internal standard obtained during the same chromatographic run. In a new approach employed in this work, in order to maximize the reliability and consistency of normalization across a large number of samples and complex sample media, internal standard signals were each normalized

by their own mean signals (throughout the entire analysis period) to yield an indicator of self-normalized instrument sensitivity. Analyte  signal was then normalized by the mean self-normalized responses of the three closest internal standard species. This approach has multiple benefits. First, the responses of sample or external standard compounds are not artificially deflated or inflated due to their proximity to internal standard compounds that have higher or lower sensitivities based on their functional





groups and derivatization. Second, this approach enables inclusion and utilization of incomplete data; in previous approaches,
if an internal standard cannot be recovered in every sample it cannot be used for normalization, as this would create
inconsistencies for the species that are otherwise assigned to that compound. Compounds at the very high and very low ends
of the volatility space are chemically important but detectable at baseline low levels that can drop below limits of detection
during periods of low sensitivity. Having to discard these species due to a few instances of missing corresponding internal
standard data causes losses of valuable information. Finally, this approach decreases analysis sensitivity to any errors and noise
in internal standard identification or isolation, as erroneously high or low individual internal standard responses are moderated
by averaging with the other nearby internal standard species. Volatility-based sensitivity corrections, which can be achieved
by raw internal standard normalization, were achieved in this work through normalization by an external standard-determined
response curve, as described in "Featurization and Target Selection for Quantification Modelling."

## 3      Data Preparation and Featurization

The analytical pipeline for data preparation through performance evaluation of this random forest modelling work is
illustrated in Figure 1. The processes and decision making around featurization, feature selection, and target selection for both
chemical properties modelling and quantification modelling, as well as the curation of the training, test, and extrapolation data
sets, is described below.

### 3.1      Featurization, Feature Selection, and Target Selection

As the aim of this work is to develop methods that can be applied to novel species not included in mass spectral
databases, features utilized in this analysis rely solely upon the information readily available for unidentifiable species. Given
the size and complexity of the intended use data suites, features must also be automatically generatable from the instrument
data output and not rely upon any visual or manual categorization by researchers. In order to make these models more broadly
useful to the atmospheric community, less common features produced by the GCxGC-MS instrumental setup (e.g. second
dimension retention time and high resolution spectra) are not utilized for chemical properties modelling in order to increase
the method's applicability to single dimension GC-MS systems and instruments with lower resolution mass spectra.

### 3.1.1      Mass Spectral Featurization

The only chemical information directly produced by GCxGC-MS for unidentified organic species are their locations
in GCxGC volatility-polarity space and mass spectra. These sources of information are therefore exclusively utilized in
creating and selecting the features for chemical properties modelling. The retention index of each compound was directly
utilized as a feature, but the mass spectra require interpretation in order to be used.

The unit mass resolution spectra utilized in this analysis include each charged fragment represented by its measured
mass to charge ratio (*m/z*) and a relative signal score out of 1000 (normalized by the most abundant fragment's peak signal).





EI is a high energy or "hard" ionization technique which typically leaves only a small fraction of molecular ions intact and creates positively charged ion fragments that are almost all singly charged, with any multiply charged ions at extremely low abundance. This means that the molecular formulae cannot generally be directly determined from the mass spectrum, even when the spectra are high resolution, and measured ions can be assumed to have a single charge. That said, the m/z of charged fragment ions yield useful information into chemical characteristics and functional groupings that can provide critical chemical information; for example, a peak at $m/z$=73 corresponds to a fragment of $Si(CH_3)_3^+$, a derivatization fragment which indicates that the ionized compound contained an OH group which was derivatized (see section 2.3). The mass differences between charged peaks also represent important pieces of information, as they can indicate losses of uncharged molecular fragments that similarly point to the structure and characteristics of the original compound. It is important to note that not all neutral mass differences between charged peaks can be interpreted as direct neutral losses as not all high m/z charged fragments directly fragment onto lower m/z charged fragments in a manner that can be directly interpreted from neutrally charged fragment losses. However, frequently occurring neutral differences may still hold value in reflecting a common coordination of neutral loss processes.

The greatest chemical information lies in features that exist in an intermediate range of occurrence frequency in the data set. A feature which appears in all the training species does not provide any useful information in predicting properties of the test species. Neither does a feature which is totally unique to a single species, as it does not provide any information on patterns which can be used to adjust prediction of properties for other species. This logic can be applied to mass spectral featurization; while it would be possible to convert every m/z to a feature and so input the entire raw mass spectrum of each compound as a series of features for the random forest model, this approach would be inefficient, open to error introduced by noise, and miss the important information provided by neutral mass differences between charged fragments.

Multiple approaches for mass spectral featurization were tested to optimize the number of features and representation of features. Given the final choice in model structure (random forest, as described in section 4), inclusion of covarying features or more features than necessary did not introduce significant sources of error. Target-specific feature restriction based on importance is discussed in section 4. The final mass spectral featurization method selected for this analysis, a simplified adaptation of methodology described in Eghbaldar et al., 1998, was as follows: the top 5 charged fragments (mass spectral peaks) from each training set mass spectrum are selected. The mass differences between these 5 peaks (a maximum of 10 numbers, if all fragments occur at differently spaced $m/z$) were then compiled into a list of "neutral losses". The charged fragment lists and the neutral loss lists of all training set external standard compounds were next combined in a frequency list, with each charged fragment or neutral loss quantified by frequency (how many compounds in the external standard test set exhibited that charged fragment or neutral loss among their top 5 peaks). The top 40 most common charged fragments and top 20 most common neutral losses were converted into features. The identities of these 40 most common fragments and 20 most common neutral mass differences (along with possible identities and notes) can be found in tables A2 and A3, respectively. The mass spectra of all training, test, and extrapolation set compounds were then simplified using the previously described method (top 5 peaks extracted and mass differences between those peaks calculated). Each $m/z$ feature was assigned the


normalized signal of that peak in the mass spectrum if the feature *m/z* was one of the top 5 peaks; otherwise, it was set to zero. Each neutral loss feature was assigned true or false for each compound depending on whether the neutral loss appeared in the
mass differences between the 5 most significant peaks. An example mass spectral featurization for the example compound hexadecane can be found in Table A4, and the mass spectral featurization process is included in the open-source R script accompanying this publication.

### 3.1.2    Target Selection for Chemical Properties Modelling

The goal of chemical properties modelling is to enable inclusion of " unidentified" species in aerosol organic analysis
that has previously been restricted to species for which the identity or at least chemical formula is known. One way in which complex organic mixtures are visualized and analyzed is through orientation of observed species in chemical properties spaces that have been developed and broadly utilized in the field of aerosol science. Two such spaces include the Volatility Basis Set (VBS (Donahue et al., 2006)) and the visualization by average carbon oxidation state and carbon number developed in Kroll et al., 2011, hereafter referred to as $\overline{OS_c}$-$n_c$ space. Compounds can be plotted in VBS space by their O:C or $\overline{OS_c}$ (average carbon
oxidation state(Kroll et al., 2011)) against some measure of volatility, either log(Vapor Pressure) or log($C_0$), where $C_0$ is the pure component sub-cooled liquid vapor pressure in atm. In $\overline{OS_c}$-$n_c$ space, compounds are plotted by their average carbon oxidation state ($\overline{OS_c}$) against carbon number. The ability to map novel or unidentifiable compounds in these spaces would provide critical information about the properties of the individual species, enable identification of groups of chemically distinct novel compounds deserving particular consideration, and more completely visualize the distribution of chemical characteristics
for complex mixtures and potential routes of chemical transformation (e.g. oligomerization, functionalization, fragmentation) beyond the identifiable components. With these goals in mind, the properties selected to be the targets of these modelling efforts were number of carbons ($n_c$), O:C, $\overline{OS_c}$, and vapor pressure.

Carbon number, O:C, and $\overline{OS_c}$ (based on the equation in Kroll et al., 2011) can all be directly calculated from chemical formula, which was known for each standard and ambient extrapolation compound (see section 3.2). Vapor pressure is not
directly calculatable from chemical formula and not all identified compounds in the external standard and extrapolation data sets have reliable experimental vapor pressure measurements available, so structurally-based vapor pressure predictions are utilized instead. Isaacman-Vanwertz and Aumont, 2021 finds that of all structure-based vapor pressure prediction methods available, the average of predictions generated by the EVAPORATION (Compernolle et al., 2011), Nannoolal (Nannoolal et al., 2008) , and Simpol (Pankow and Asher, 2008) models yields the most accurate vapor pressure prediction. These methods
were therefore utilized to predict the vapor pressures of all standard and extrapolation set compounds, and the average structurally predicted vapor pressures were utilized as the "true" vapor pressures for model training and evaluation. Seven of the external standard test set species and fifteen of the extrapolation set species were incompatible with the prediction capabilities of one or more of the three structural vapor pressure prediction methods and were therefore not utilized in performance analysis. Two additional potential targets, double bond equivalent and H:C ratio, were tested but failed to produce
sufficiently robust property predictions.



The final components of the chemical properties random forest models are as follows:

Targets: Carbon number, $\overline{OS_c}$, O:C, vapor pressure (structurally modelled)

Features: Retention index, 40 feature representation of mass spectral charged fragments, 20 feature representation of neutral mass differences between charged fragments

325   A table listing the entire set of input features for chemical properties modelling of the example compound Hexadecane can be found in table A4, and the instrument-produced mass spectrum for this species can be found in Figure A1.

### 3.1.3   Featurization and Target Selection for Quantification Modelling

Compound quantification factor is significantly and reliably related to retention index across all compound classes tracked, but this relationship is not linear and changes much more rapidly in some retention index windows than others. This

phenomenon, caused by incomplete cold inlet trapping of species in the most volatile sensitivity region and incomplete thermal desorption of species in the least volatile sensitivity region, is illustrated in Figure A2 and is consistent with findings presented in Zhang et al., 2018. A variety of retention index corrections were tested, including the following: a) factorizing the retention indices of each compound (rounded to the nearest 100) and including as a feature in model training and testing, and b) normalizing (dividing) each compound by the raw signal of its nearest deuterated alkane internal standard, the method utilized

in Zhang et al., 2018. Both methods however performed poorly in the 1600-1900 RI range, where response increases extremely rapidly with RI (Figure A2). The most reliable normalization method and the method selected for this analysis was normalizing (dividing) all compound quantification factors by the average response curve for alkanes, defined by the combination of 2 best fit exponential curves, which intersect at RI ≈ 1950 as illustrated in Figure A2, and training on/predicting this normalized response factor rather than the raw quantification factor. The $r^2$ of the exponential fit of individual calibration period

quantification factors around the response curve in the volatile region is .77, while the $r^2$ of the curve describing the less volatile region is .65. Note that these fits take into account each quantification factor of each calibration window, and are therefore influenced by the variations in the measured quantification factors of the same compounds measured at different points throughout analysis. RI-normalized response factors were translated back to predicted quantification factors for performance evaluation, as other methods of quantification do not utilize this normalization method.

Unlike in the case of chemical properties modelling, quantification modelling performance was significantly improved by inclusion of second dimension retention time information, and it was therefore included as a feature in response factor prediction. As a result, this approach in its current form is only usable by GCxGC-MS applications, but could be adapted to single dimension chromatography-mass spec.

In this analysis, continuous measurement periods (consecutively collected samples) were analysed in sequences

bounded by calibration curve runs. To preserve the quantification continuity in these consecutive measurements and avoid step changes in calculated concentration that might occur due to switching between quantification factors, the two quantification factors bookending an analysis period are averaged to assign the quantification factors for samples run in that interval. To replicate this approach, the compound quantification factors were sequentially averaged to yield 5 quantification periods (the


final calibration curve experienced an instrument failure, and the last calibration period is therefore based solely on the final curve).

The mass spectral featurization is described in "Mass Spectral Featurization" above.

The final components of the quantification model are as follows:

Target: Normalized response factor (RI curve-normalized, calibration period averaged)

Features: Retention index, second dimension retention time, calibration period, 40 feature representation of mass
spectral charged fragments, 20 feature representation of neutral mass differences between charged fragments

### 3.2 Training, Test, and Extrapolation Set Curation

To generate a training and test set from the external standard data, each external standard was assigned to a chemical group (alkane, sugar, PAH, etc), and the list of external standard compounds was randomly split 80:20 (80% of compounds in the training set, 20% in the test set) maintaining the ratios of different chemical groups. 200 possible splits were generated,
and the split which demonstrated the greatest similarity in median retention index and median second dimension retention time between the test and training sets was selected to avoid potential extrapolation problems that might occur with a highly skewed distribution of test and training compounds across the GCxGC space. This process is documented in Supporting Information.

The extrapolation set was curated from the compounds isolated from the GoAmazon samples by comparing the spectra and retention indices of compounds to the external standard and matches in the NIST14 mass spectral database. Of the ~1500
unique compounds identified across 12 template samples, 63 were determined to match external standard compounds and an additional 71 compounds were identifiable from the NIST library due to high (>800, (Worton et al., 2017)) mass spectral match factor and retention index agreement with database entries. Based on number of silicon atoms in the assigned formulae from the NIST identification, each chemical formula was converted to its underivatized form. Only the 71 compounds that were identifiable from the NIST library but not from external standards were included in the extrapolation set to ensure that
performance metrics for the extrapolation set would not be skewed by the inclusion of species that may have been in the training data, and to ensure that the test set and extrapolation set performance evaluations would be entirely independent. As illustrated in Figure 2, the distribution of training, test, and extrapolation set species utilized in this work effectively span the distribution of unknown compounds in GCxGC volatility-polarity space.

### 4 Model Selection, Training, and Tuning

The number and complexity of input features and lack of clear linear relationships between target properties and input features in this analysis is well suited to a decision tree-based analytical approach (Bentéjac et al., 2021; Rokach, 2016). Random forest and gradient boosting methods were both preliminarily tested for response factor prediction. Random forests demonstrated slightly better performance and was selected for this and additional methodological reasons, as follows. Random Forests are more robust to overfitting than gradient boosting, which is a particular concern in this case given the small number



of training compounds (~100) compared to the large numbers of novel environmental organics that are the intended subjects of unverifiable modelling. Additionally, random forests perform well using the default settings and do not require extensive tuning to achieve optimal performance (Bentéjac et al., 2021). As the aim of this work is to produce models that the atmospheric science community, including non-experts in machine learning, can easily implement for novel compound analysis, this robustness and simplicity is a significant advantage.

The training and tuning processes for chemical properties prediction are visualized in Figure 1. For each target property, the model was trained on the external standard training set data, the curation of which is described above. As previously referenced, random forests do not require extensive tuning, and for ease of use reasons most parameters were maintained at their default values. Tuning primarily focused on feature restriction. Feature restriction to enforce tree diversity (mtry) was optimized by 5-fold cross validation, with the mtry value that minimized mean average error (MAE) selected. Although random

forest modelling is comparatively not influenced by the inclusion of features that do not contribute significant predictive capabilities, the inclusion of unnecessary features can contribute to overfitting of the training data which decreases prediction performance for the test and extrapolation data sets. To address this problem, the feature importance (a measure of increase in node purity when this feature is used in a split) of each input feature was extracted from the original predictive model. The importance metrics were normalized by the total importance of all features to generate a percent importance score for each

feature. Importance distributions were highly skewed, with a relatively low number of features contributing the majority of decrease in node purity. Features that contributed less than 1% to the total importance score were removed, and the model was re-trained on only the important features. Extrapolation set performance improvements from removal of low importance features was low, with an improvement in $OSR^2$ (out of sample $R^2$, defined in detail in section 5.1) on the order of 0-0.03. This indicates that this step is not crucial for chemical properties or quantification factor prediction. The cross validation-optimized

mtry number, number of important features, and identity of important features for the chemical properties models (one optimized model per property predicted) are summarised in Table 1. For quantification modelling, mtry is optimized at 44 features and 46 features meet an importance criterion of >1 %.

## 5    Model Performance Evaluation

### 5.1    Chemical Properties Modelling Performance

Three performance metrics are utilized to evaluate target predictions for the four chemical properties models. The first, out-of-sample $r^2$ ($OSR^2$), provides a measure of how significantly a model improves upon a baseline assumption that all target property values are equal to the mean of those values in the training data. It approaches a maximum of 1 for perfect predictions. The second metric, mean average error (MAE) provides the mean absolute prediction residual in the units of the target property. This metric is particularly important, as it provides a benchmark for prediction accuracy which can be translated into

visualization and utilized to determine which applications are appropriate given prediction errors. The final performance



metric, root mean square error (RMSE), is also a scale dependent error metric and provides the quadratic mean of prediction residuals. The equations for these metrics are provided below:

$$OSR^2 = 1 - \frac{\sum_{i=1}^{n}(T_i - P_i)^2}{\sum_{i=1}^{n}(T_i - \bar{R}_T)^2} \qquad (1)$$


$$MAE = \frac{\sum_{i=1}^{n}|P_i - T_i|}{n} \qquad (2)$$

$$RMSE = \sqrt{\frac{\sum_{i=1}^{n}(P_i - T_i)^2}{n}} \qquad (3)$$

In this notation, for each test or extrapolation set compound i summed across a population of n compounds, $T_i$ indicates the true value of the property being tested, $P_i$ indicates the predicted value of that property, and $\bar{R}_T$ indicates the mean of the

selected property in the training data set.

The prediction performance for the tuned and trained chemical properties model are evaluated independently on both the external standard test set (Figure 3, Table 2) and the ambient sample extrapolation set (Figure 4, Table 3). Both of these performance evaluations are important for different reasons. The external standard contains many series of highly chemically similar species (for example alkane and carboxylic acid series), meaning that the test set is likely to be more chemically similar

to the training set than a real distribution of ambient organic species would be. Performance evaluation on the extrapolation set therefore provides a more realistic assessment of likely prediction accuracies on the large number of novel ambient organic compounds that are the intended focus of this modelling effort. That said, prediction performance on the external standard test set also yields important information. The external standard is designed to cover the entire space of anticipated chemical features for the environmental samples and is therefore more diverse relative to the number of compounds included compared

to the extrapolation set (which is primarily CHO-type compounds). Performance evaluation on the external standard test set therefore yields more information about model performance across a broad suite of compound classes.

### 5.1.1 Test Set Performance Evaluation

By all evaluation metrics applied (summarised in Table 2), performance for carbon number, O:C, carbon oxidation state, and log(VP) predictions on the external standard test set are robust. The O:C and carbon number predictions are

particularly strong, with OSR$^2$ of .89 and .88 respectively and average errors of .072 element ratio units and 1.8 carbon number units. For context, given the range in true values from O:C= 0-1 and carbon number = 4-31, both mean average errors are approximately 7% of the range of measured values. For $\overline{OS_c}$ and vapor pressure, the average errors normalized by the measurement range are both approximately 12%. As illustrated Figure 3, this means that the distribution of predicted properties usefully and reliably reflects the distribution of true properties and indicates that the random forest-based model provides





useful information that allows a wide range of compound classes to be reliably characterized based on mass spectrum and
retention index.

### 5.1.2    Extrapolation Set Performance Evaluation

As discussed above, while the external standard test set provides useful information on model performance across a
wide range of compound types, its performance is potentially inflated by high degree of chemical similarity between the
training and test set compounds. Performance evaluation on the ambient sample extrapolation set is therefore likely a more
accurate indicator of prediction performance on novel or uncatalogued species. Of the four properties modelled, the
performances for carbon number prediction and carbon oxidation state remain consistent or slightly improve (carbon number
$OSR^2$ increases to .93), while O:C and log(VP) prediction performances decrease, both in terms of $OSR^2$ and MAE.

The weakest extrapolation set performance by far is vapor pressure prediction, which drops to an $OSR^2$ of .68. The
correlation between predicted and true properties is also the weakest (as illustrated in Figure 4), with particularly large
prediction residuals for the highest volatility species. For example, the extrapolation set compound with the highest vapor
pressure prediction error is 1,2-Benzenedicarboxylic acid, which has a retention index of <1400 making it more volatile than
the most volatile internal standard compound. While this compound does not lie outside of the volatility and polarity boundaries
of the external standards in GCxGC space, is significantly more volatile than any diacid compound in the standard mixture,
and the influence of double derivitization on its true ambient volatility relative to the chromatographic elution time of its
derivatized form may not have been appropriately captured. Unlike the other properties targeted in this analysis, vapor pressure
is not directly calculable based on chemical formula and poses challenges for many techniques; as discussed in Isaacman-
Vanwertz and Aumont, 2021, molecular structure plays an important role in volatility, which significantly limits the accuracy
with which techniques that identify formula but not structure (typically chemical ionization techniques) can predict the true
volatility of their measured components. A more complete comparison between the random forest model's performance in
vapor pressure prediction compared to other techniques used throughout the field is therefore required to provide context for
vapor pressure prediction errors in the ambient sample extrapolation set (further discussed below in section 5.1.3).

For both O:C and $\overline{OS_c}$ (which are highly related properties), extrapolation set prediction performance suffers at the high
end of the oxygenation scale, although the performance reduction is far more pronounced for O:C prediction. This is due to
the lack of highly oxygenated species in the external standard; random forest models do not extrapolate beyond the range of
properties in the training data and therefore cannot predict O:C ratios of higher than 1.5 when that is above the maximum in
the training data. The extraneously highly oxidized species for which O:C and $\overline{OS_c}$ prediction accuracy suffers lie almost
exclusively in the most volatile region instrument sensitivity, where vapor pressure prediction inaccuracies have been
previously described. As a result, extrapolation set property prediction for O:C, $\overline{OS_c}$, and log(VP) were restricted to compounds
above a retention index of 1500. As illustrated in Figure A3 and Figure 2, the significant majority of ambient analytes were
above the 1500 retention index threshold, justifying the decision to restrict prediction of these properties to the retention index


window in which their performance is better optimized. In applying these techniques to the larger suite of novel species, maintaining these retention window restrictions is critical to avoid the introduction of significant sources of error.

Given the strong and consistent performance of carbon number and $\overline{OS_c}$ predictions across the majority of the retention
index space and between both test and extrapolation sets, the most robust visualization of chemical properties based on random forest predictions is likely to be in $\overline{OS_c}$-$n_c$ space (Kroll et al., 2011). Predicting the carbon numbers and $\overline{OS_c}$ of the known ambient compounds and superimposing the true and predicted property distributions in the $\overline{OS_c}$-$n_c$ space highlights the strengths and weaknesses of chemical properties modelling.  To better represent the prediction capabilities of the full chemical space and the scope of information that would be provided for properties prediction on a complex sample including hundreds
of individual species, all identifiable ambient compounds (including those that overlap with the external standard) were included in property prediction and visualization. As illustrated in Figure 5, the real and predicted chemical properties spaces for the ambient data set indicate both strengths and weaknesses for this application of chemically properties modelling. As noted earlier, random forest modelling does not extrapolate and has a tendency to underpredict property extremes. This is apparent in both the high $\overline{OS_c}$ region and the high carbon number regions of the $\overline{OS_c}$-$n_c$ space, where high carbon oxidation
states and high carbon numbers were both underpredicted. These errors could be moderated by adding more oxygenated species and higher carbon number species to the external standard, which would provide the model with more information to predict properties in these regions. In a context of extended continuity of analysis of similar sample media, this suggests an iterative approach in which the addition of new standards to a calibration mixture can be prioritized through analysing the chemical features of poorly predicted compounds in the sample media and adding new standards that replicate those features. Despite
the prediction errors visualized in Figure 5, the overall shape of the true chemical properties space was extremely well represented by predictions. While conclusions based on the presence or absence of extremes in predicted properties would not be appropriate, analyses based on the relative distributions of populations of interest provides valuable insight comparable to other parameterizations of compound properties from incomplete knowledge.

### 5.1.3    Vapor Pressure Modelling: Comparison to Prior Methods

Chromatography using a non-polar column is intended to separate compounds by volatility and has been used to directly predict novel compound vapor pressures in previous studies (Isaacman-VanWertz et al., 2016). It is therefore important in this context to evaluate both how significantly random forest modelling improves upon simple linear modelling of volatility based on retention index as well as how this method compares to other parameterizations of vapor pressure. As illustrated in Figure 6 and Table 4, the log(VP) prediction residuals for random forest model predictions indicate that random forest-generated
predictions are both more accurate and more precise than predictions by the linearized retention index method or from the Li et al., 2016 chemical formula-based parameterization, as they demonstrate a tighter distribution that is more centered around zero. The mean absolute error for random forest vapor pressure prediction is significantly lower than errors from both predictions based on retention index (t-test p value = .01) and predictions based on chemical formula (t-test p value = $3.1 \times 10^{-5}$).





**5.2    Quantification Modelling Performance**

The approach for evaluating performance for quantification modeling requires slight alterations compared to property prediction. Although the random forest model predicts the residuals of quantification factors around the retention index response normalization curve (Figure A2) rather than directly, these residuals are converted back to quantification factors for both the true and predicted properties for performance evaluation. This serves two purposes; first, other quantification methods

do not use this retention index-based normalization so conversion to absolute prediction errors is necessary to compare methods, and second, a direct quantification error assessment provides more useful and applicable information about how significantly quantification errors could influence conclusions based on model-quantified data.

The test set compounds were quantified using two alternative quantification methods, *Manual* or *Closest proxy quantification* (described in Liang et al., 2021, which utilizes a combination of both)*,* to benchmark random forest model

performance. *Manual proxy quantification* entails manually assigning a compound to a chemically similar external standard based on researcher judgement on what chemical class the unidentified compound would likely belong to based on some combination of location in GCxGC space and mass spectrum. This is the current preferred method for quantification of compounds that are not in the external standard and in theory should provide the most reliable results in cases where an extremely chemically similar standard is available, but it is highly inefficient and relies upon researcher judgement calls which

are difficult to standardize. *Closest proxy quantification* assigns each compound to its nearest external standard in GCxGC space, or to an average of the nearest standards within a set radius limitation. In this work, the average of the quantification factors of the 3 nearest standard species was used, as this demonstrated improved performance compared to single closest proxy quantification. This method is efficient, but it introduces potentially significant error by assigning species with different chemical characteristics (and therefore different quantification factors) the same response factor if they are sufficiently close

in GCxGC space. Each test set compound was assigned to a proxy quantification factor from the training set based on each of these two methods, and each proxy compound's quantification factor at each time point was substituted as a prediction of the test set compound's quantification factor at that calibration window.

The standard performance metrics for quantification factor prediction using the random forest model, manual proxy quantification and closest proxy quantification, are compared in table 5. The random forest model significantly outperforms

both other methods; it has a relatively high $OSR^2$ of .65 compared to negative $OSR^2$ values for the two proxy methods (indicating that at least on average, assuming all test compounds have the same quantification factor as the average of all training set compounds would have performed better than proxy quantification). MAE and RMSE also indicate improved performance when using the random forest model over other methods. While these metrics provide useful information on model performance, they do not reveal why the performance (particularly of the proxy methods) is so poor and do not provide

useful information to evaluate likely propagation of quantification errors. Unlike for the chemical properties modelling, for quantification modelling % error is a much more important metric than absolute error, because it translates directly to how significant total quantification error across a large suite of compounds is likely to be and provides insights into underlying





biases in different methods. Figure 7 illustrates the quantification factor % error distributions of the three methods and demonstrates the improved performance of random forest modelled quantification predictions on three criteria. First, as illustrated by panel A, random forest modeling produces far fewer and less extreme outlier prediction errors that are orders of magnitude different from the true values. These result when a compound that the instrument is extremely insensitive to (which would have a true extremely low quantification factor) is assigned a moderate or high quantification factor. In practice the influence of these types of quantification inaccuracies is very limited as few ambient species that the instrument is this significantly insensitive to would occur above detection limits, but they could introduce errors nonetheless. Here it is important to keep in mind that each point represents a single quantification from a single calibration period; some outliers therefore indicate compounds that exhibited extremes in quantification factors during a single calibration period. This was most common among standard compounds at the edges of the instrument's sensitivity window, as these species are more significantly impacted by alterations in instrument performance. Second, as illustrated by Figure 7 panel B, the error distribution for the random forest model is significantly more centered around zero compared to either proxy model. Median random forest model quantification error is -2%, compared to 17% for closest proxy quantification and 19% for manual proxy quantification. In practice, this indicates that over a large number of quantified species, random forest modeling is unlikely to introduce biased quantifications that might skew results, while the two proxy methods would likely inflate the apparent mass of novel compounds. Third, also illustrated by Figure 7 panel B (though less directly), random forest modeling produces prediction errors more tightly distributed around the median, meaning that the absolute % error distribution for random forest modelling also outperforms the two proxy methods. Median absolute % error for random forest model predictions is 37%, compared to 57% for the closest proxy method and 41% for the manually assigned proxy method. The average % error improvements from random forest modeling compared to both proxy methods are statistically significant (t-test p values both < .0004), but the median absolute % error distributions of the random forest and manually assigned proxy quantifications are not significantly different based on a Mood's median test. The random forest and closest proxy method absolute % error distribution differences are statistically significant, with a Mood's test p value of .001. While critical for contextualizing the potential impact of quantification errors on mass attribution of complex mixtures, a % error-based analysis of prediction accuracy is necessarily asymmetrical, as a predicted quantification factor can produce a minimum of -100 % error (the case if the predicted value were to be zero) but far more than +100% error if the quantification factor is significantly overpredicted. A symmetrical error analysis of log(predicted quantification factor/true quantification factor), illustrated in Figure A4, is required to probe the frequency and dynamics of underprediction in greater depth. Figure A4 demonstrates that the random forest model is more prone to underprediction outliers, but continues to outperform the other methods in achieving a narrow error distribution centered at zero.

A final benefit of random forest modeling-based quantification not captured in the performance metrics is the ability to utilize incomplete data. With proxy quantification, any standard compound that cannot be calibrated for at any point over the course of an analysis cannot be used, as the species that are calibrated by that compound would not be quantifiable during the window with missing calibration data. The random forest-based quantification method relies upon the entire external





standard suite to inform corrections for instrument performance over time and can therefore produce robust quantification factor predictions even when individual standard calibration curves are missing for a particular calibration window. This allows for significantly greater flexibility in analysis, as compounds can be added to the external standard if they are observed in
initial samples and still be usable to inform quantifications for periods before they were present.

In summary, random forest quantification factor modelling significantly outperforms both closest proxy and manual proxy quantification methods. It is significantly more efficient than manual proxy modeling, exhibits fewer outliers of multi order of magnitude overestimations, produces an error distribution that is more centered around zero (preventing significant biases in total mass over large numbers of quantified and summed species), and exhibits improvements in absolute percent
error of predictions.

## 6    Conclusions

This work presents a new machine-learning based method for quantifying and predicting chemical properties of novel organic compounds observed in the atmosphere. Based on a relatively small combined training and test set of ~130 known compounds, we are able to predict the carbon numbers, vapor pressures, carbon oxidation states, and O to C ratios of ambient
organic compounds with sufficient accuracy to usefully represent compound distributions in chemical property spaces that are important in atmospheric science. That these predictions are generated solely from retention indices and unit mass resolution mass spectra marks a significant step forward in ability to characterize the novel organic components of earth's atmosphere based on measurements generated from a wide range of commonly available atmospheric instrumentation. In GCxGC-MS applications, these methods contribute significant improvements in both accuracy and analytical efficiency for novel
compound quantification that enable users to perform untargeted analysis of the rich complexity of data generated by advances in instrumentation. While the untargeted analysis data science techniques described in this work have been developed for and tested on atmospheric applications, they are not structurally limited in scope and could be applied to a wide range of chromatographic-mass spectral data sets to enable characterization of complex organic mixtures. The open-source R script published in supplement to this work is intended to provide a framework for groups throughout the atmospheric chemistry
community to efficiently apply and adapt these methods to broadly enhance our ability to take advantage of the increasingly complex information provided by ever accelerating advances in environmental chemistry instrumentation.

### Code Availability

Sample code designed for adaptation and use by other users is available in the GitHub repository associated with this manuscript (https://github.com/ebarnesey/Ch3MS-RF, https://doi.org/10.5281/zenodo.6320122). The knit R markdown
including primary analysis is included in "Supporting Information."



**Data Availability**

Composition and metadata related to the external standard is available in the GitHub repository associated with this manuscript (https://github.com/ebarnesey/Ch3MS-RF, https://doi.org/10.5281/zenodo.6320122). Mass spectra and unique identifiers of

species from the ambient samples collected during the GoAmazon field campaign are available through the Goldstein Library of Biogenic and Environmental Spectra (UCB-GLOBES). All raw data can be provided by the corresponding authors upon request.

**Author Contributions**

AG and LY planned the measurements. EF prepared and analyzed the standards and samples. LY and RW supervised data

collection and preliminary analysis. PG supervised development and selection of data analysis and modeling techniques. BA contributed modeled compound property data. EF wrote the manuscript draft; LY, BA, PG, and AG reviewed and edited the manuscript.

**Competing Interests**

The authors declare no conflict of interest.

**Acknowledgements**

We gratefully acknowledge the support of the National Science Foundation Graduate Research Fellowship Program (DGE-1752814), as well as the Department of Energy Atmospheric System Research program (DE-SC0020051). The described work was conducted under DOE-ASR support and benefits from samples collected at the GoAmazon field campaign, a DOE-ASR supported observational campaign. Paul Grigas's research is supported in part by NSF AI Institute for Advances

in Optimization Award 211253 and NSF Award CMMI-1762744.

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





**FIGURES**

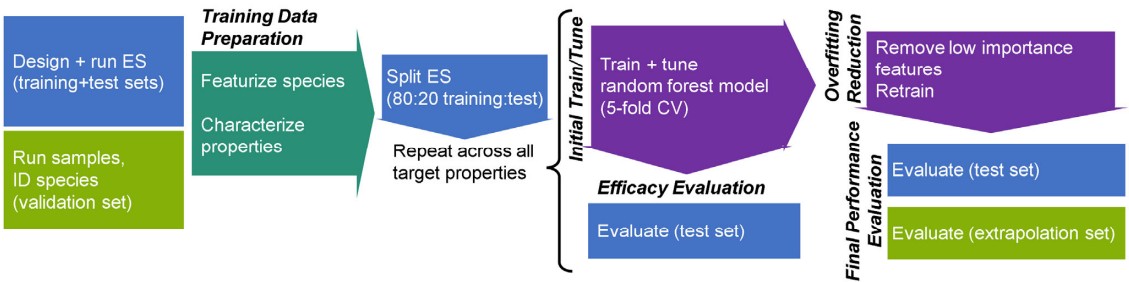

**Figure 1: Analytical pipeline for chemical properties modelling using a random forest model. ES indicates external standard; CV indicates cross validation**

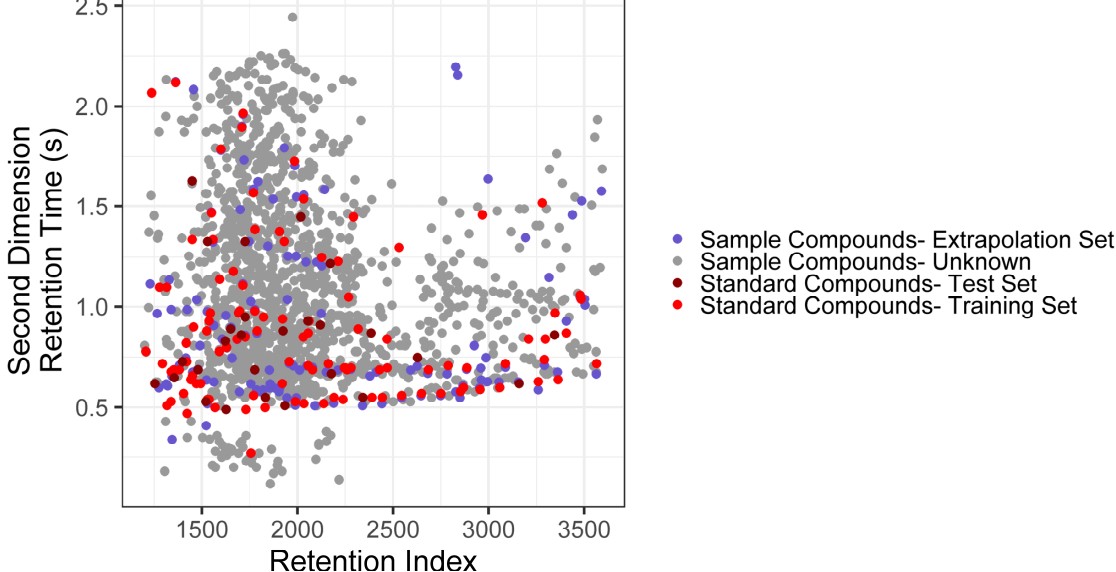

**Figure 2: Distribution of training, test, extrapolation, and unidentified sample compounds in two-dimensional chromatographic chemical properties space**






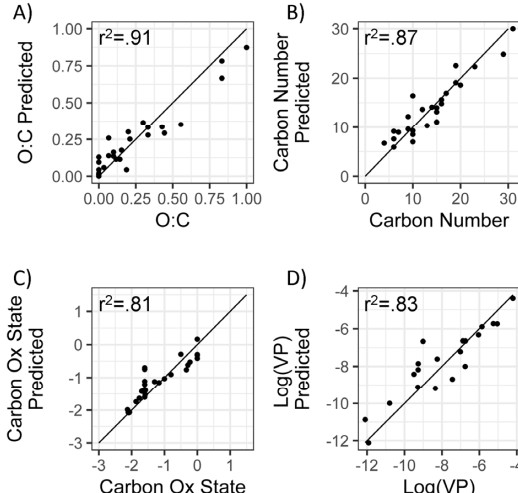

**Figure 3: External standard test set true and predicted chemical properties from random forest modelling**

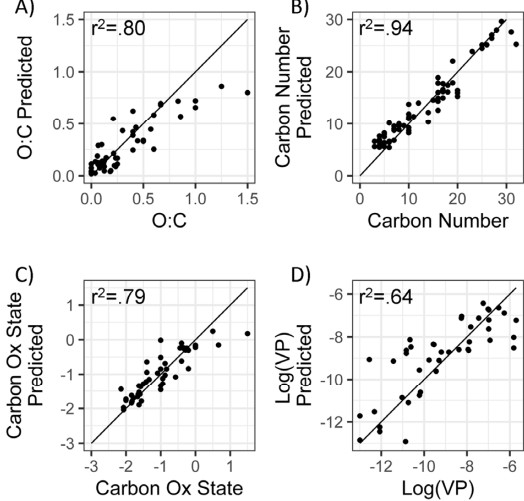

**Figure 4: Ambient extrapolation set true and predicted chemical properties from random forest modelling**





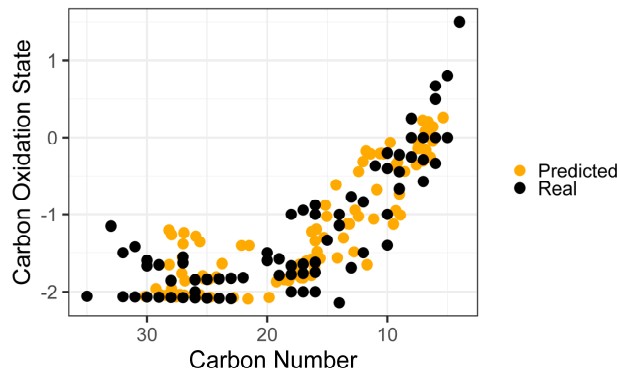

**Figure 5**: **True versus predicted chemical properties distribution of ambient sample organic species within a Carbon Number v. Carbon Oxidation State space**

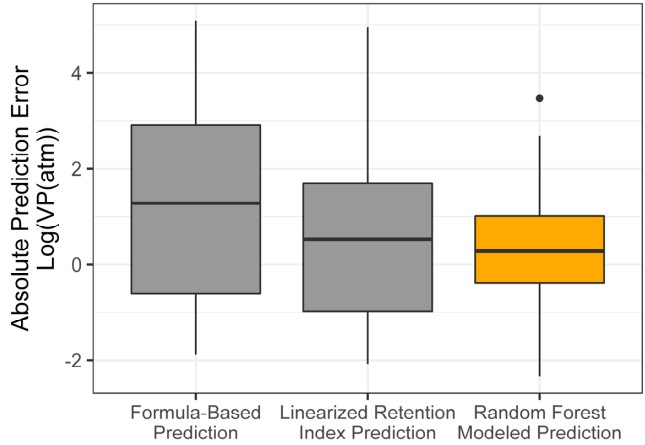

**Figure 6: Vapor pressure prediction residuals (Log(VP), VP in atm) for vapor pressure predictions of the ambient extrapolation set based on formula-based parameterization (Li et al., 2016), linearized retention index-based modelling, and random forest modelling.**





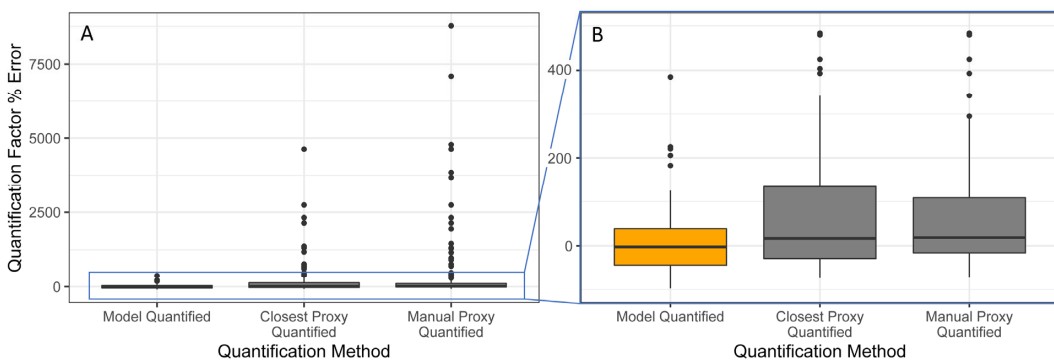

**Figure 7: Quantification performance comparison between random forest model (orange) and two previously utilized quantification methods, specifically closest proxy quantification and manually assigned proxy quantification. Midline of boxes indicates sample median, while top and bottom indicate 25th and 75th percentiles. Linear "whiskers" extend to the least extreme values within 1.5 × the inner quartile range of the sample. Disconnected dots indicate sample outliers that fall beyond the whisker parameters.**





**TABLES**

**Table 1: Tuning parameters and important features for chemical properties prediction models. m/z indicates charged fragment features and n indicates neutral mass difference features**

| Property Model | Optimized mtry | Number of Important Features | Important Features |
|---|---|---|---|
| O:C | 4 | 19 | Retention index, m/z 41, m/z 43, m/z 45, m/z 57, m/z 69, m/z 73, m/z 74, m/z 75, m/z 103, m/z 113, m/z 147, m/z 189, m/z 204, m/z 217, n 2, n 15, n 28, n 30 |
| Carbon Number | 6 | 9 | Retention index, m/z 41, m/z 45, m/z 55, m/z 57, m/z 73, m/z 99, n 14 |
| Average Carbon Oxidation State | 4 | 17 | Retention index, m/z 41, m/z 43, m/z 45, m/z 55, m/z 57, m/z 69, m/z 73, m/z 75, m/z 91, m/z 93, m/z 117, m/z 119, m/z 147, n 1, n 2, n 30 |
| Log(Vapor Pressure) | 9 | 9 | Retention index, m/z 55, m/z 73, m/z 75, m/z 129, m/z 145, m/z 147, n 3, n 30 |

**Table 2: Performance metrics for random forest-based modelling of chemical properties of the external standard test set. "Range of true properties" units in units of property: O:C in unitless atom#/atom#, Carbon Number in atom#, average carbon oxidation state in mean charge, and Log(Vapor Pressure) in Log(atm).**

| Property | Out of Sample $R^2$ | Mean Average Error | Root Mean Square Error | Range of True Properties |
|---|---|---|---|---|
| O:C | .89 | .072 | .094 | 0-1 |
| Carbon Number | .88 | 1.8 | 2.4 | 4-31 |
| Average Carbon Oxidation State | .79 | .24 | .33 | (-2.1)- 0 |
| Log(Vapor Pressure) | .82 | .72 | .93 | (-12)-(-4.2) |

**Table 3: Performance metrics for random forest-based modelling of chemical properties of the ambient aerosol sample extrapolation set. "Range of true properties" units in units of property: O:C in unitless atom#/atom#, Carbon Number in atom#, average carbon oxidation state in mean charge, and Log(Vapor Pressure) in Log(atm).**

| Property | Out of Sample $R^2$ | Mean Average Error | Root Mean Square Error | Range of True Properties |
|---|---|---|---|---|
| O:C* | .78 | .11 | .17 | 0-1.5 |
| Carbon Number | .93 | 1.8 | 2.2 | 3-32 |
| Average Carbon Oxidation State* | .80 | .25 | .37 | (-2.1)- (1.5) |
| Log(Vapor Pressure)* | .68 | 1.0 | 1.4 | (-13)- (-5.7) |

**\*Restricted to retention index > 1500**



**Table 4. Error distribution metrics random forest model, retention index linear model, and formula-based predictions of vapor pressure. All reported errors in units of log(VP(atm)).**

| Vapor Pressure Prediction Method | Mean Error | Median Error | Mean Absolute Error | Median Absolute Error |
|---|---|---|---|---|
| **Random Forest Model** | .24 | .21 | 1.1 | .76 |
| Retention Index Linear Model | .55 | .52 | 1.5 | 1.1 |
| Formula-Based Parameterization | 1.2 | 1.3 | 2.0 | 1.3 |

**Table 5. Performance metrics for quantification factor prediction for three methods of unidentified compound quantification: random forest modelling, manually assigned proxy quantified, and closest proxy quantified.**

| Quantification Method | Out of Sample $R^2$ | Mean Average Error | Root Mean Square Error |
|---|---|---|---|
| **Random Forest Model** | .65 | .00085 | .0021 |
| Manually Assigned Proxy Quantified | -4.1 | .0036 | .0080 |
| Closest Proxy Quantified | -1.8 | .0026 | .0059 |










**Appendix A: Supplementary Tables and Figures**

FIGURES

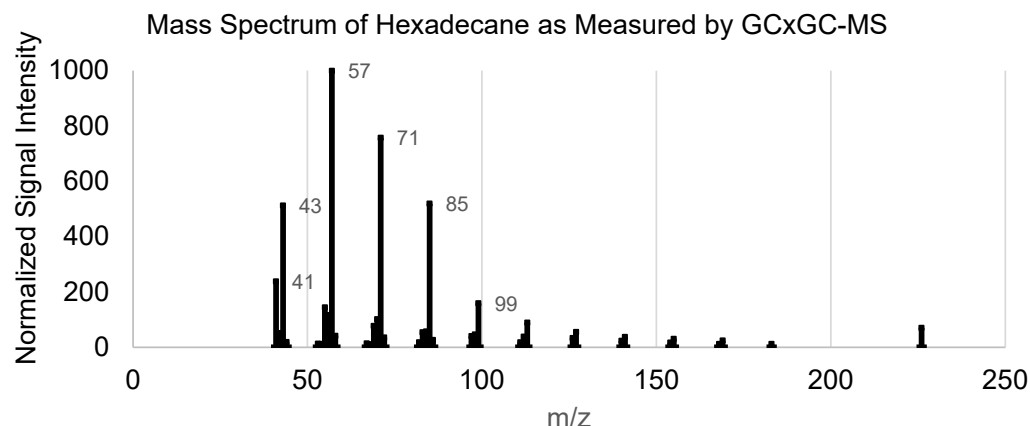

**Figure A1: Mass Spectrum of Hexadecane as measured by GCxGC-MS and featurized in table A2.**

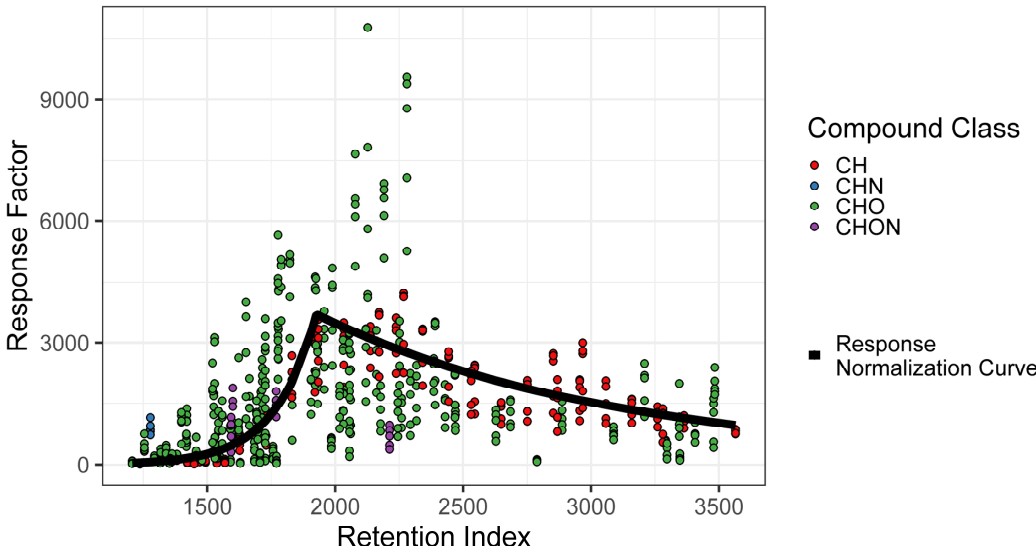

**Figure A2: Quantification factor normalization curve based on average response factors of alkanes**



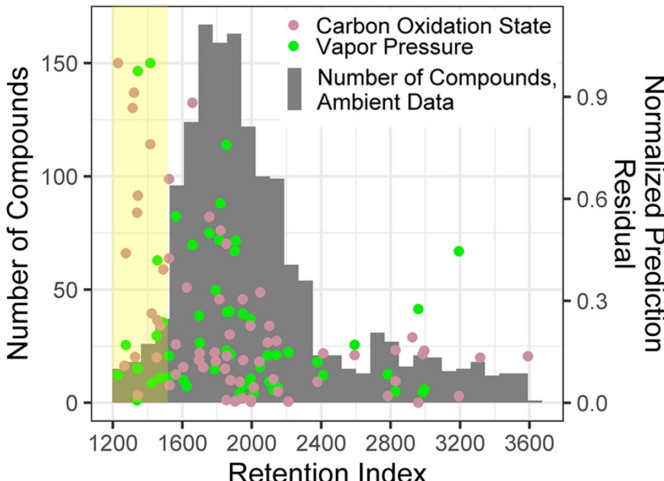

**Figure A3: Normalized prediction residuals of carbon oxidation state and vapor pressure v. retention index for ambient data compound property predictions set, overlaid with compound number distribution over the retention index for ambient data set. The yellow highlighted region indicates compounds below a retention index of 1500.**

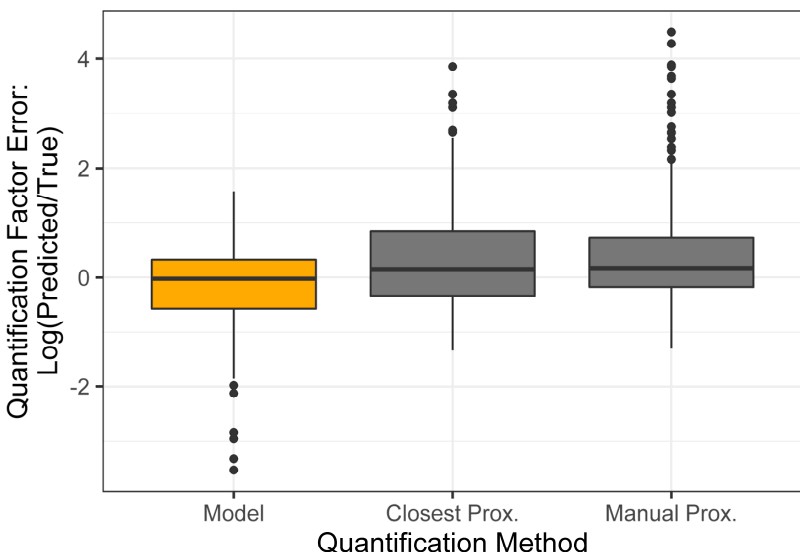

**Figure A4: Quantification factor prediction errors expressed in Log(predicted quantification factor/true quantification factor) for test set quantification factors predicted by random forest model (orange), closest proxy, and manual proxy methods.**




TABLES

**Table A1: External standard names, formulae (underivatized), retention indexes, split (training set versus test set), and manually assigned quantification proxies.**

| Name | Chemical Formula | Retention Index* | Split | Manual Proxy |
|---|---|---|---|---|
| 12-OH C18 acid | C18H36O3 | 2470 | Train | |
| 16-OH C16 acid | C16H32O3 | 2429 | Train | |
| 2-ketoglutaric acid | C5H6O5 | 1629 | Train | |
| 3-5-dimethoxyphenol | C8H10O3 | 1525 | Train | |
| 4, 4 dimethoxy-benzophenone | C15H14O3 | 2293 | Train | |
| 4-hydroxybenzoic acid | C7H6O3 | 1651 | Test | 2-ketoglutaric acid |
| 4-nitrocatechol | C6H5NO4 | 1769 | Train | |
| 4-terpineol | C10H18O | 1206 | Train | |
| 9H-florenone | C13H8O | 1778 | Train | |
| a-amyrin | C30H50O | 3479 | Train | |
| abietic acid | C20H30O2 | 2468 | Train | |
| anthraquinone | C14H8O2 | 2017 | Test | xanthone |
| benzophenone | C13H10O | 1664 | Train | |
| beta-caryophyllene aldehyde | C15H24O2 | 1715 | Train | |
| beta-caryophyllinic acid | C14H22O4 | 2060 | Train | |
| beta-caryophyllonic acid | C15H24O3 | 1931 | Train | |
| beta-nocaryophyllinic acid | C13H20O5 | 2127 | Train | |
| beta-nocaryophyllone aldehyde | C14H22O3 | 1757 | Train | |
| beta-nocaryophyllonic acid | C14H22O4 | 1985 | Train | |
| beta-sitosterol | C29H50O | 3406 | Train | |
| bisabolol | C15H26O | 1770 | Train | |
| borneol | C10H18O | 1254 | Test | nonanol |
| C10 carboxylic acid | C10H20O2 | 1479 | Test | dimethyl glutaric acid |
| C10 diacid (sebacic acid) | C10H18O4 | 1922 | Train | |
| C12 diacid | C12H22O4 | 2120 | Test | beta-caryophyllinic acid |
| C13 acid | C13H26O2 | 1776 | Test | vanillic acid |
| C14 alkane | C14H30 | 1422 | Train | |
| C14 diacid | C14H26O4 | 2317 | Train | |
| C16 alkane | C16H34 | 1626 | Train | |
| C16 acid | C16H32O2 | 2078 | Train | |
| C17 alkane | C17H36 | 1730 | Test | C17 alkane |
| C17 acid | C17H34O2 | 2177 | Test | linoleic acid |
| C18 alkane | C18H38 | 1830 | Train | |
| C18 acid | C18H36O2 | 2280 | Train | |





| C19 alkane | C19H40 | 1934 | Test | C20 alkane |
|---|---|---|---|---|
| C20 alkane | C20H42 | 2034 | Train | |
| C21 alkane | C21H44 | 2137 | Train | |
| C22 alkane | C22H46 | 2238 | Train | |
| C22 acid | C22H44O2 | 2684 | Train | |
| C23 alkane | C23H48 | 2341 | Test | C24 alkane |
| C24 alkane | C24H50 | 2443 | Train | |
| C24 acid | C24H48O2 | 2886 | Train | |
| C25 alkane | C25H52 | 2545 | Train | |
| C26 alkane | C26H54 | 2649 | Train | |
| C26 acid | C26H52O2 | 3088 | Train | |
| C27 alkane | C27H56 | 2750 | Train | |
| C28 alkane | C28H58 | 2852 | Train | |
| C28 acid | C28H56O2 | 3291 | Train | |
| C29 alkane | C29H60 | 2955 | Train | |
| C30 alkane | C30H62 | 3058 | Train | |
| C31 alkane | C31H64 | 3159 | Test | C30 alkane |
| C32 alkane | C32H66 | 3259 | Train | |
| C33 alkane | C33H68 | 3363 | Train | |
| C35 alkane | C35H72 | 3564 | Train | |
| C7 acid | C7H14O2 | < 1400 | Train | |
| C8 acid | C8H16O2 | 1293 | Train | |
| C9 acid | C9H18O2 | 1381 | Train | |
| C9 diacid (azelaic acid) | C9H16O4 | 1822 | Train | |
| cholesterol | C27H46O | 3209 | Train | |
| chrysene | C18H12 | 2531 | Train | |
| cis-vaccenic acid | C18H34O2 | 2259 | Train | |
| citronellol | C10H20O | 1338 | Train | |
| cycloisolongifolene | C15H24 | 1355 | Test | pyrocatechol |
| DEET | C12H17NO | 1600 | Train | |
| deoxycholic acid | C24H40O4 | 3347 | Train | |
| dibenz(ah)anthracene | C22H14 | 3280 | Train | |
| dimethyl glutaric acid | C7H12O4 | 1456 | Train | |
| dodecyl benzene | C18H30 | 1920 | Train | |
| eicosanol | C20H42O | 2390 | Train | |
| ergosterol | C28H44O | 3296 | Train | |
| erythreitol | C4H10O4 | 1528 | Train | |
| FAME16 (methyl palmitate) | C17H34O2 | 1957 | Train | |
| FAME18 (methyl stearate) | C19H38O2 | 2161 | Train | |





| farnesol | C15H26O | 1832 | Test | bisabolol |
|---|---|---|---|---|
| galactosan | C6H10O5 | 1684 | Train | |
| gamma dodecalactone | C12H22O2 | 1709 | Train | |
| glyceric acid | C3H6O4 | 1352 | Train | |
| hexadecanamide | C16H33NO | 2212 | Train | |
| hexadecanol | C16H34O | 1989 | Train | |
| homosalate | C16H22O3 | 2054 | Test | beta-caryophyllinic acid |
| hydroquinone | C6H6O2 | 1420 | Train | |
| ionone | C13H20O | 1449 | Train | |
| isoeugenol | C10H12O2 | 1591 | Train | |
| isopimaric acid | C20H30O2 | 2385 | Test | C14 Diacid |
| ketopinic acid | C10H14O3 | 1530 | Test | pinonic acid |
| levoglucosan | C6H10O5 | 1726 | Train | |
| linoleic acid | C18H32O2 | 2245 | Train | |
| lupeol | C30H50O | 3483 | Train | |
| maltol | C6H6O3 | 1316 | Train | |
| mannosan | C6H10O5 | 1706 | Test | galactosan |
| MBTCA | C8H12O6 | 1776 | Train | |
| Me-OH-glutatric acid | C6H10O5 | 1623 | Test | 2-ketoglutaric acid |
| monopalmitin | C19H38O4 | 2628 | Test | monostearin |
| monostearin | C21H42O4 | 2788 | Train | |
| nonanol | C9H20O | 1318 | Train | |
| octadecanal | C18H36O | 2056 | Train | |
| octadecanol | C18H38O | 2191 | Train | |
| octadecanone | C18H36O | 2031 | Train | |
| oleic acid | C18H34O2 | 2251 | Train | |
| palmitoleic acid | C16H30O2 | 2056 | Train | |
| p-anisic acid (4-methoxybenzoic acid) | C8H8O3 | 1544 | Train | |
| pentadecanone | C15H30O | 1726 | Test | pinic acid, isomer 1 |
| perylene | C20H12 | 2967 | Train | |
| phthalic acid | C8H6O4 | 1714 | Train | |
| phthalimide | C8H5NO2 | 1593 | Train | |
| pinic acid, isomer 1 | C9H14O4 | 1692 | Train | |
| pinic acid, isomer 2 | C9H14O4 | 1697 | Train | |
| pinonic acid | C10H16O3 | 1550 | Test | hexadecanamide |
| pyrene | C10H16 | 2171 | Train | |
| pyrocatechol | C6H6O2 | 1339 | Train | |
| quinoline | C9H7N | 1278 | Train | |



| resorcinol | C6H6O2 | 1399 | Test | hydroquinone |
|---|---|---|---|---|
| retene | C18H18 | 2267 | Train | |
| Sesquiterpene 1[†] | C15H24 | 1404 | Train | |
| Sesquiterpene 2[†] | C15H24 | 1442 | Test | Sesquiterpene 3 |
| Sesquiterpene 3[†] | C15H24 | 1449 | Train | |
| Sesquiterpene 4[†] | C15H24 | 1451 | Train | |
| Sesquiterpene 5[†] | C15H24 | 1471 | Train | |
| Sesquiterpene 6[†] | C15H24 | 1493 | Train | |
| Sesquiterpene 7[†] | C15H24 | 1537 | Train | |
| Sesquiterpene 8[†] | C15H24 | 1569 | Train | |
| Sesquiterpene 9[†] | C15H24 | 1610 | Train | |
| sinapinaldehyde | C11H12O4 | 2032 | Train | |
| squalene | C30H50 | 2868 | Train | |
| stigmasterol | C29H48O | 3344 | Test | ergosterol |
| syringaldehyde | C9H10O4 | 1726 | Test | 9H-florenone |
| syringic acid | C9H10O5 | 1924 | Test | C10 diacid (sebacic acid) |
| syringol | C8H10O3 | 1418 | Train | |
| threitol | C4H10O4 | 1521 | Test | erythreitol |
| triacetin | C9H14O6 | 1362 | Train | |
| tridecanal | C13H26O | 1537 | Train | |
| vanillic acid | C8H8O4 | 1789 | Train | |
| vanillin | C8H8O3 | 1558 | Train | |
| verbenone (-) | C10H14O | 1237 | Train | |
| xanthone | C13H8O2 | 1906 | Train | |

*Normalized by deuterated alkane standard series

[†]Isomer identity undetermined, only quantification factor and properties related to chemical formula included in modelling

**Table A2: 40 most common charged fragments featurized for mass spectral featurization, with possible formulae and implications of published peaks.**

| Fragment m/z | Possible Formulae | Notes |
|---|---|---|
| 41 | C3H5+ | |
| 43 | C3H7+, C2H3O+ | Propyl group, ketone indicator |
| 45 | CHO2 | Carboxyl indicator, underivitized |
| 55 | | |
| 56 | | |
| 57 | $C_4H_9$+, C3H5O+ | Signature alkane fragment, ketone/ester |
| 67 | | |
| 69 | | |
| 71 | C4H7O+ | Ketone/ester |
| 73 | $Si(CH_3)_3$+ | Indicates derivatization and therefore presence of OH group |





| 74 | | |
|---|---|---|
| 75 | | |
| 77 | C6H5+ | phenyl |
| 79 | | |
| 81 | | |
| 83 | | |
| 85 | | |
| 91 | | |
| 92 | | |
| 93 | C6H5O+ | Oxygenated aromatics |
| 95 | | |
| 99 | | |
| 103 | | |
| 105 | | |
| 107 | | |
| 109 | | |
| 111 | | |
| 113 | | |
| 117 | | |
| 119 | | |
| 121 | | |
| 129 | | |
| 131 | | |
| 132 | | |
| 135 | | |
| 145 | | |
| 147 | | |
| 189 | | |
| 204 | Si2C8H20O2+ | Indicative of sugars |
| 217 | Si2C9H21O2+ | Indicative of sugars |

**Table A3: 20 most common neutral mass differences between charged peaks, selected for mass spectral featurization, with possible formulae and implications of commonly reported neutral losses.**

| Neutral Loss/Mass Difference (amu) | Probable Formulae/ Interpretation | Notes |
|---|---|---|
| 1 | Loss of H | |
| 2 | | |
| 3 | | |
| 4 | | |
| 6 | | |
| 8 | | |
| 10 | | |
| 11 | | |
| 12 | | |
| 13 | | |
| 14 | | |
| 15 | CH3 | Methyl |
| 16 | O | Alcohol- derivatization agent loss |





| 18 | | |
|----|----|----|
| 20 | | |
| 26 | | |
| 27 | | |
| 28 | CO | Carbonyl |
| 30 | | |
| 42 | | |

**Table A4. Full chemical properties modelling features for Hexadecane**

| Feature | Feature Class | Feature Input |
|---------|---------------|---------------|
| Retention Index (d-alkane normalized) | Chromatography | 1627 |
| m/z 41 | Mass Spectrum Common Fragment | 238 |
| m/z 43 | Mass Spectrum Common Fragment | 512 |
| m/z 45 | Mass Spectrum Common Fragment | 0 |
| m/z 55 | Mass Spectrum Common Fragment | 144 |
| m/z 56 | Mass Spectrum Common Fragment | 116 |
| m/z 57 | Mass Spectrum Common Fragment | 999 |
| m/z 67 | Mass Spectrum Common Fragment | 0 |
| m/z 69 | Mass Spectrum Common Fragment | 0 |
| m/z 71 | Mass Spectrum Common Fragment | 757 |
| m/z 73 | Mass Spectrum Common Fragment | 0 |
| m/z 74 | Mass Spectrum Common Fragment | 0 |
| m/z 75 | Mass Spectrum Common Fragment | 0 |
| m/z 77 | Mass Spectrum Common Fragment | 0 |
| m/z 79 | Mass Spectrum Common Fragment | 0 |
| m/z 81 | Mass Spectrum Common Fragment | 0 |
| m/z 83 | Mass Spectrum Common Fragment | 0 |
| m/z 85 | Mass Spectrum Common Fragment | 519 |
| m/z 91 | Mass Spectrum Common Fragment | 0 |
| m/z 92 | Mass Spectrum Common Fragment | 0 |
| m/z 93 | Mass Spectrum Common Fragment | 0 |
| m/z 95 | Mass Spectrum Common Fragment | 0 |
| m/z 99 | Mass Spectrum Common Fragment | 0 |
| m/z 103 | Mass Spectrum Common Fragment | 0 |
| m/z 105 | Mass Spectrum Common Fragment | 0 |
| m/z 107 | Mass Spectrum Common Fragment | 0 |
| m/z 109 | Mass Spectrum Common Fragment | 0 |
| m/z 111 | Mass Spectrum Common Fragment | 0 |
| m/z 113 | Mass Spectrum Common Fragment | 89 |
| m/z 117 | Mass Spectrum Common Fragment | 0 |
| m/z 119 | Mass Spectrum Common Fragment | 0 |
| m/z 121 | Mass Spectrum Common Fragment | 0 |
| m/z 129 | Mass Spectrum Common Fragment | 0 |
| m/z 131 | Mass Spectrum Common Fragment | 0 |
| m/z 132 | Mass Spectrum Common Fragment | 0 |
| m/z 135 | Mass Spectrum Common Fragment | 0 |





| m/z 145 | Mass Spectrum Common Fragment | 0 |
| m/z 189 | Mass Spectrum Common Fragment | 0 |
| m/z 204 | Mass Spectrum Common Fragment | 0 |
| m/z 217 | Mass Spectrum Common Fragment | 0 |
| loss of 1 | Mass Spectrum Neutral Loss/Mass Diff. | FALSE |
| loss of 2 | Mass Spectrum Neutral Loss/Mass Diff. | TRUE |
| loss of 3 | Mass Spectrum Neutral Loss/Mass Diff. | FALSE |
| loss of 4 | Mass Spectrum Neutral Loss/Mass Diff. | FALSE |
| loss of 6 | Mass Spectrum Neutral Loss/Mass Diff. | FALSE |
| loss of 8 | Mass Spectrum Neutral Loss/Mass Diff. | FALSE |
| loss of 10 | Mass Spectrum Neutral Loss/Mass Diff. | FALSE |
| loss of 11 | Mass Spectrum Neutral Loss/Mass Diff. | FALSE |
| loss of 12 | Mass Spectrum Neutral Loss/Mass Diff. | FALSE |
| loss of 13 | Mass Spectrum Neutral Loss/Mass Diff. | FALSE |
| loss of 14 | Mass Spectrum Neutral Loss/Mass Diff. | TRUE |
| loss of 15 | Mass Spectrum Neutral Loss/Mass Diff. | FALSE |
| loss of 16 | Mass Spectrum Neutral Loss/Mass Diff. | TRUE |
| loss of 18 | Mass Spectrum Neutral Loss/Mass Diff. | FALSE |
| loss of 20 | Mass Spectrum Neutral Loss/Mass Diff. | FALSE |
| loss of 26 | Mass Spectrum Neutral Loss/Mass Diff. | FALSE |
| loss of 27 | Mass Spectrum Neutral Loss/Mass Diff. | FALSE |
| loss of 28 | Mass Spectrum Neutral Loss/Mass Diff. | TRUE |
| loss of 30 | Mass Spectrum Neutral Loss/Mass Diff. | TRUE |
| loss of 42 | Mass Spectrum Neutral Loss/Mass Diff. | FALSE |