# Peer review of "Ch3MS-RF: A Random Forest Model for Chemical Characterization and Improved Quantification of Unidentified Atmospheric Organics Detected by Chromatography-Mass Spectrometry Techniques"

_Atmospheric Measurement Techniques, 2022_

## Author Comment (AC1)

Responses to Reviewers: Ch3MS-RF

Reviewer 1

*This work presents a new method for predicting organic molecule properties (carbon number, mean oxidation state, oxygen-to-carbon ratio, vapor pressure) for compounds measured by gas chromatography and mass spectrometry but not listed in mass spectral databases. The novel idea is to use a statistical model trained on compounds listed in such databases together with parameters extracted from measurement, which is available regardless of the compound being listed in such a database. The authors include the caveat that this approach requires two-dimensional gas chromatography measurements that capture both volatility and polarity and on dimension is insufficient, which does require more complicated instrumentation than commonly deployed. Nonetheless, the general idea can be useful for the atmospheric science community, and is recommended for publication in Atmospheric Measurement Techniques. I note a few comments regarding the generalizability below.*

We thank the reviewer for this helpful summary and these comments. We do wish to clarify that two-dimensional gas chromatography is only required for quantification factor prediction which utilized second dimension retention information, as all chemical properties were well predicted using exclusively first dimension retention index (equivalent to standard GC-MS retention index) and mass spectrum.  We clarify this point in lines 636 to 642 of newly added section 5.3, *Considerations for Adaptation Across Instruments and Methods.*

*The authors refer to their "extrapolation set" in they are not included in the training set, but in reality it appears that new samples span a subset of the feature domain spanned by the training set. The consideration of whether extrapolation in this sense is happening or likely to happen in new data sets is relevant because random forest is not capable of such extrapolations - and this would limit the model's utility substantially. Can the authors clarify this point?*

We thank the reviewer for this very helpful note.  As noted, random forest modelling does not effectively extrapolate beyond the feature space of the training data set, and as such our terminology was somewhat unclear. We added the following text to section 3.2 *Training, test, and extrapolation set curation* (lines 376-385) to clarify that the extrapolation set is intended to test whether the training set is sufficiently similar to the sample medium of interest to indicate that the model will be capable of appropriately modelling the target compound properties, as follows:

> "The methodology described in this work cannot effectively extrapolate beyond the feature space of the training data set, and the identifiable organic compounds in the Amazonian aerosol samples are defined as an "extrapolation set" not because they test the abilities of the model to extrapolate beyond the feature space boundaries of the external standard training data, but because they represent the true range of individual isomer-specific identities observed in ambient samples. These compounds test the model's ability to extrapolate property prediction beyond the compound groups included in the external standard and indicate whether the sample is sufficiently similar to the training data to make this approach appropriate for the target sample medium, as extremely high prediction inaccuracies indicate compound classes too dissimilar from the training data to be appropriately modelled using Ch3MS-RF."

I assume the results are solely applicable to samples run on the same instrument with the same protocol, as the retention time is dependent on the operating procedure. For any new protocol a new model would have to be trained. Can this model be used to generate predictions using measurements on similar instruments using the same protocol, or does a new model have to be trained on each instrument? For publication, the authors should include a statement regarding what is required for adaptation by other users.

Thank you for this helpful comment. The protocol utilized in this methodology normalizes retention times to an alkane internal standard series such that retention indices are indicative of when compounds elute relative to known compounds (normal alkanes). Using a retention index is a very standard technique in chromatography to compare results for samples run on different instruments. In cases where elution times are normalized to known compounds such as normal alkanes, and similar phase columns are utilized, this technique is highly adaptable across instruments and techniques. To more quantitatively address this point, we have added section 5.3 *Considerations for Adaptation Across Instruments and Methods* and figure 8, in which we test how sensitive prediction performances for each property are to drifts in retention indices. The results of this analysis indicate that O:C and average carbon oxidation state predictions are not significantly affected by retention time drifts between the training set and the test set, while carbon number and vapor pressure predictions are more sensitive but still robust within drifts the equivalent of up to 1 carbon number unit. Section 5.3 now addresses this and other concerns, highlighting that spectra and retention times/indices produced by other instruments may be used, so long as retention index drifts can be normalized to less than the equivalent of the elution time between two linear alkanes difference, the column type is standardized, and the retention times and spectra of oxygenated species are consistently either derivatized or underivatized.

**5.3 Considerations for Adaptation Across Instruments and Methods**

[revised manuscript text omitted]

---

## Author Comment (AC2)

Responses to Reviewers: Ch3MS-RF

Reviewer 2

*This study developed new machine learning techniques to characterize unidentifiable organic compounds using GC-MS and GCxGC-MS techniques. The authors provided a detailed discussion and demonstration of this model and its potential to improve the current understanding of undefined organic species in the atmosphere. This new method is able to improve the quantification accuracy compare with manual proxy modeling, which will lead to a better understanding of atmospheric organic aerosols formation and chemical properties. I'm supportive of this paper and recommend for publication in Atmospheric Measurement Techniques.*

*Here are a few minor comments:*

*Authors have mentioned that the vapor pressures were calculated for model training and evaluation, a few external standards test set and extrapolation set species were incompatible with vapor pressure prediction, can the author provide more explain more about how it is incompatible and the evaluation process for vapor pressure? Based on Figure 4, it seems like the predicted vapor pressure has more variability than other perimeters, and the more accurate vapor pressure can improve the model accuracy.*

We thank reviewer 2 for this opportunity to clarify the reasoning behind restricting the vapor pressure training and prediction datasets. The Nanoolal and SIMPOL vapor pressure estimates were generated using the GECKO-A tool, which in its current form does not allow certain molecular structures (eg PAHs) or functional groups (e.g. amines) to be processed. While some of these species could have been predicted by hand, manual vapor pressure predictions were not feasible at the number scale of compounds utilized in this analysis. The referenced version of EVAPORATION does not contain parameterizations for some functional groups such as amines and heterocyclic compounds, rendering these species outside the scope of estimates using these methods. Under ideal circumstances, validated experimental vapor pressures for all test and training compounds would be used to produce the most accurate predictions possible, but experimental vapor pressures are not available for many more species than were excluded due to lack of structurally predicted vapor pressures and other compounds have conflicting experimental vapor pressures reported in the literature. The method described in this work was selected for a few reasons. First, there is a high overlap between compounds whose vapor pressures could be predicted by SIMPOL, EVAPORATION, and Nannoolal and compounds in our training test and validation sets. Second, the referenced work in Isaacman-Van Wertz and Aumont et al., 2021 finds that the average of these methods produces optimally accurate vapor pressure predictions for the compound classes most commonly observed in the extrapolation set. Third, the structural vapor pressure predictions were both efficient and internally consistent, particularly when compared to compiling often conflicting values from the literature. The following language has been added (line 318) to clarify the limitations which rendered some compounds incompatible with predictions.

"Seven of the external standard test set species and fifteen of the extrapolation set species were incompatible with the prediction capabilities of one or more of the three structural vapor pressure

prediction methods (most frequently due to functional group types for which the models are not parameterized) and were therefore not utilized in performance analysis."

*Author mentioned that the model underestimated the high carbon oxidation state region and the high carbon number region, but there is no predicted data shown in the plot? it also seems like the model is a little bit overpredicted for the carbon number region between 20 and 30, can author comment on that?*

Thank you so much for this very helpful note- we intended to state that the high carbon oxidation state compounds (which have low carbon numbers) have underpredicted carbon oxidation states, and that the high carbon number compounds (which have low carbon oxidation states) have underpredicted carbon oxidation states- however this was unclear and we see how this may have been interpreted as referring to compounds that have both high carbon number and high carbon oxidation state, of which there were none. The wording has now been altered (line 498) to clarify that these are independent observations and not descriptive of any group of compounds.

> "This is apparent in both the high $\overline{OS_c}$ region and the high carbon number regions of the $\overline{OS_c}$-$n_c$ space, where high carbon oxidation states and high carbon numbers were each independently underpredicted."

For the compounds in the region between carbon number 20 and 30, while the distribution of predicted values in this region is higher than observed, this effect is primarily caused by the underprediction of carbon numbers, as now clarified in the description of Figure 5. This is to say, the predicted points are shifted to the right of the true properties positions for those compounds rather than shifted up. We believe that the clarification regarding underprediction of carbon numbers for high carbon number species now clarifies this point.

> "As illustrated in Figure 5, the real and predicted chemical properties spaces for the ambient data set indicate both strengths and weaknesses for this application of chemically properties modelling. As noted earlier, random forest modelling does not extrapolate and has a tendency to underpredict property extremes. This is apparent in both the high $\overline{OS_c}$ region and the high carbon number regions of the $\overline{OS_c}$-$n_c$ space, where high carbon oxidation states and high carbon numbers were each independently underpredicted. These errors could be moderated by adding more oxygenated species and higher carbon number species to the external standard, which would provide the model with more information to predict properties in these regions."

*Does this model capable of any GC-MS system or is there any specific requirement for the instrument? Can author add some discussion of the limitation of this model as well?*

Thank you for this comment, which is highly aligned with similar questions from Reviewer 1. To better address appropriate considerations in adapting this technique to different instruments and for incorporating data produced by multiple instruments and techniques, we have now added section 5.3 *Considerations for Adaptation Across Instruments and Methods* and figure 8, reproduced below.

**5.3 "Considerations for Adaptation Across Instruments and Methods**

[revised manuscript text omitted]